# Nanoparticle-Mediated Delivery of Flavonoids: Impact on Proinflammatory Cytokine Production: A Systematic Review

**DOI:** 10.3390/biom13071158

**Published:** 2023-07-21

**Authors:** Jazmín Cristina Stevens Barrón, Christian Chapa González, Emilio Álvarez Parrilla, Laura Alejandra De la Rosa

**Affiliations:** 1Instituto de Ciencias Biomédicas, Universidad Autónoma de Ciudad Juárez, Ciudad Juárez 32310, Mexico; jazmin.stevens@uacj.mx (J.C.S.B.); ealvarez@uacj.mx (E.Á.P.); 2Instituto de Ingeniería y Tecnología, Universidad Autónoma de Ciudad Juárez, Ciudad Juárez 32310, Mexico

**Keywords:** flavonoid, nanoparticle, drug delivery, inflammation, release, cytokine, encapsulation, bioavailability, in vitro, in vivo

## Abstract

Flavonoids are a diverse group of plant-derived compounds that have been shown to have various health benefits, including anti-inflammatory effects. However, their use in the treatment of inflammatory diseases has been limited due to their low bioavailability. The nanoparticle-mediated delivery of flavonoids has been proposed as a potential solution to this issue, as it allows the sustained release of the flavonoids over time. There are several different nanoparticle systems that have been developed for flavonoid delivery, including polymeric nanoparticles, liposomes, and inorganic nanoparticles. This systematic review aims to evaluate the impact of nanoparticle-mediated delivery of flavonoids on pro-inflammatory cytokine production in various diseases. We analyzed the performance of flavonoid-encapsulated nanoparticles in regulating cytokine production in different in vitro and in vivo studies. To this end, we followed the guidelines of the Preferred Reporting Items for Systematic Reviews and Meta-Analyses (PRISMA) to conduct a comprehensive search of the literature and to assess the quality of the included studies. The results showed that flavonoid-encapsulated nanoparticles significantly downregulated pro-inflammatory cytokines, such as TNF-α, IL-1β, IL-6, and IL-18. In some cases, this effect was significantly greater than that observed with non-encapsulated flavonoids These findings suggest that nanoparticle-mediated delivery of flavonoids may have potential as a therapeutic approach for the treatment of inflammatory diseases.

## 1. Introduction

Immune-mediated inflammatory diseases (IMIDs) are a group of conditions that affect a significant portion of the developed world’s population and are characterized by an inflammatory response [1,2]. These diseases have been linked to the development of various conditions [3], including cancer [4], obesity [5], neurodegenerative diseases [6], coronary artery disease [7], tuberculosis [8], autoimmune disorders [9], atherosclerosis [10], inflammatory bowel disease [11], psoriasis [12,13,14], rheumatoid arthritis [15], spondyloarthritis [16], etc. Cytokines are produced and secreted by immune system cells and play a pivotal role in immune-mediated inflammatory diseases [17,18,19,20,21]. Some cytokines, such as interleukin-1 (IL-1) and tumor necrosis factor (TNF), have proinflammatory effects and can contribute to the pathogenesis of these diseases. Others, such as interleukin-4 (IL-4) and interleukin-10 (IL-10), have anti-inflammatory effects and can help regulate the inflammatory response. The role of cytokines in immune-mediated inflammatory diseases is complex and depends on the specific cytokines and the specific situation (Figure 1). Current treatments for IMIDs are often ineffective as these diseases can present as heterogeneous syndromes rather than well-defined entities.

To address those issues, it is necessary to consider the immunological and molecular mechanisms underlying these diseases rather than solely focusing on their clinical symptoms. The widespread consumption of unhealthy diets, which are low in nutrients such as nuts, seeds, whole grains, and milk and high in processed meat, sugary beverages, and salt, contribute to a significant portion of diet-related deaths and disability-adjusted life years. The bioavailability of dietary compounds, including phytochemicals, is influenced by factors such as their release from food, stability in the digestive system, and ability to pass through the intestinal wall [22]. The chemical structure of phenolic compounds can also affect their absorption and the metabolites present in the bloodstream.

Phenolic compounds are a large and diverse group of compounds that include flavonoids [23,24] and other types of compounds such as tannins, lignans, and stilbenes [25]. Flavonoids are characterized by the presence of a specific chemical structure called a flavonoid skeleton and are known for their antioxidant properties. They are found in a wide variety of plant-based foods and are often associated with the colors and flavors of these foods. Many flavonoids, including quercetin, catechin, and kaempferol, have been shown to have potential health benefits including reducing the risk of cardiovascular disease and certain types of cancer [26]. In the same sense, flavonoids have been shown to have anti-inflammatory properties, which means they may help to reduce inflammation in the body. Consuming foods that are rich in flavonoids may help to reduce chronic inflammation and improve overall health. The mechanisms by which these compounds exert their anti-inflammatory effects are not fully understood, but it is thought that they may inhibit the production of pro-inflammatory cytokines, scavenge free radicals, and modulate the activity of immune cells [27,28,29,30].

Nanotechnology-based delivery systems may improve the effectiveness of dietary compounds, such as phenolic compounds [31,32,33]. This could provide new nutritional approaches and personalized diets using these formulations as drug-delivery carriers for phenolic compounds. The solubility of phenolic compounds plays a crucial role in their bioavailability and, consequently, their efficacy. However, when phenolic compounds are loaded into nanoparticles, it is the physicochemical properties of the nanoparticles themselves that primarily determine the efficacy of the formulation. Factors such as the size, shape, and surface properties of the nanoparticles become key determinants of the formulation’s effectiveness. Additionally, after the incorporation of phenolic compounds into a specific nanocarrier, the stability assessment of the formulation is based on its overall composition and characteristics. It is through careful consideration and optimization of these factors that the desired efficacy and functionality of the phenolic compound-loaded nanoparticles can be achieved. In general, the encapsulation efficiency of a compound in a nanoparticle refers to the amount of the compound that is successfully incorporated into the nanoparticle, which can affect its bioavailability and efficacy. With this in mind, we have formulated the following research question: What is the effect of flavonoid-encapsulated nanoparticles on the production of pro-inflammatory cytokines? We also highlight some of the challenges and limitations of using nanoparticle systems for flavonoid delivery and suggest directions for future research in this area.

In the literature, there are several studies on flavonoids released by nanoparticles. The systematic review conducted by Ghanbari-Movahed et al. [34] aimed to evaluate the preventive and therapeutic effects of naringin against human malignancies. The review identified some challenges in utilizing naringin as a therapeutic agent, including the limited in vivo studies and the need to determine the appropriate dose range for clinical application. Furthermore, the review emphasized the importance of taxonomic validation of the material under investigation and the development of novel delivery systems, such as surface-modified naringin-nanostructures. Moreover, the meta-analysis conducted by Xie et al. [35] demonstrated that flavonoid supplementation in animals exposed to nanomaterials can enhance antioxidative enzyme levels, reduce oxidative mediators, alleviate cell apoptosis and DNA damage, and mitigate organ injuries. These findings support the potential of nanoparticles to enhance the effect of flavonoids providing a rationale for further research in this area. However, no available study shows the relationship between flavonoids encapsulated in nanoparticles and the production of proinflammatory cytokines. The objective of this review is to describe the impact of flavonoid-encapsulated nanoparticles on the production of pro-inflammatory cytokines.

## 2. Materials and Methods

This review was conducted in accordance with the PRISMA systematic review statement guidelines. A comprehensive search of PubMed, ScienceDirect, and EBSCO host was conducted using the search strategy “(Flavonoid) AND (nanoparticles) AND (cytokines) AND (in vitro OR in vivo)”. The inclusion criteria for this systematic review included studies published in the English language within the past 7 years that evaluated the effect of flavonoids encapsulated or loaded in nanoparticles on the secretion of pro-inflammatory cytokines in in vitro or in vivo studies. The studies were also required to report on the flavonoid encapsulation efficiency, indicate the composition of the nanoparticles, report on the physicochemical properties of the nanoparticles such as size and zeta potential, and indicate the percentage of release at 24 h or that could be determined with the results. In addition, the studies were required to indicate or show the effect of encapsulation on interleukin inhibition. The exclusion criteria for this systematic review included reports of antioxidant molecules other than flavonoids, flavonoids that were not encapsulated with nanoparticles, and flavonoids encapsulated with nanoparticles in which no pro-inflammatory cytokines were quantified.

### Inclusion and Eligibility Criteria

In this systematic review, we employed specific inclusion and eligibility criteria to identify relevant studies investigating the impact of nanoparticle-mediated delivery of flavonoids on proinflammatory cytokine production. The selected studies were original research articles published in peer-reviewed journals and focused on evaluating the effects of various nanoparticle delivery systems on proinflammatory cytokine levels. Both in vitro and in vivo studies were included, utilizing relevant disease models. The primary outcome of interest was the assessment of proinflammatory cytokines such as TNF-α, IL-1β, IL-6, and IL-18, and the comparison between flavonoid-encapsulated nanoparticles and non-encapsulated flavonoids or control groups. Language restrictions were applied, and studies published in the English language were included.

We conducted a thorough search and reviewed the studies that met our criteria. The systematic review followed a specific selection process, which was reported using a PRISMA flow diagram. The identified studies were imported into Mendeley, where duplicate records were removed manually. Two reviewers independently screened the titles and abstracts of the studies against the eligibility criteria and removed those that did not meet the criteria. The remaining studies were retrieved in full for further assessment by the reviewers, and any that were deemed ineligible were removed. OHAT Rob was used in the review to assess the risk of bias in the included studies.

## 3. Results

The identification process of this systematic review began by searching three databases: PubMed, Science Direct, and EBSCO host. A total of 52 records were identified in PubMed before delimiting the 7-year period; when delimiting the initial year to 2016 the number of records was 47. Similarly, 2445 records were identified from Science Direct initially, when delimiting the publication period from 2016 and before screening records with automated tools, such as choosing article type, and 40 records were identified from the EBSCO host. Before the screening process, duplicate records were removed (*n* = 4) and records marked as ineligible by automation tools were removed (*n* = 1999). Additionally, 21 records were removed because they were papers other than original publications, such as reviews, conference proceedings, etc. The remaining 513 records were screened, resulting in the exclusion of 242 records. One report was sought for retrieval and assessed for eligibility along with the remaining 271 records. The eligibility assessment resulted in the exclusion of 91 reports that focused on antioxidant molecules other than flavonoids, 86 reports that focused on flavonoids, but these were not encapsulated with nanoparticles, and 69 reports that focused on flavonoids encapsulated with nanoparticles but did not quantify proinflammatory cytokines. A total of 24 [36,37,38,39,40,41,42,43,44,45,46,47,48,49,50,51,52,53,54,55,56,57,58] studies were included in the review (Figure 2).

The 23 eligible studies (14 in vitro and 9 in vivo) fulfilled the eligibility criteria, with some minor exceptions. For example, in most of the studies, the percentage release of the flavonoid at 24 h was reported or can be determined; in only [53,55], the study time was up to 9 h and 12 h, respectively; while in [37,48], it was not possible to determine these data. Table 1 summarizes the main characteristics of the selected studies. It can be seen that the objectives of these studies are all related to the investigation of the effects of various substances, such as Au-mPEG(5000)-S-HP nanoparticles, naringenin nanoparticles, apigenin, eupafolin, chrysin-loaded PLGA-PEG nanoparticles, and curcumin-loaded nanoparticles, on inflammation and related processes in various in vivo and in vitro models. The specific aims of each study vary, but they generally involve evaluating the ability of the substances to modulate inflammation, assess their mechanisms of action, and assess their potential therapeutic benefits for the treatment of conditions characterized by inflammation. These conditions may include liver cancer, acute kidney injury, pulmonary inflammation and fibrosis, and various inflammatory diseases. The studies involve a range of methodologies, including in vivo animal studies, in vitro cell culture experiments, and molecular and biochemical assays.

The OHAT Rob tool was applied to assess the risk of bias in all included studies, including those that used cell culture models. The OHAT Rob tool is a standardized tool that aims to help researchers evaluate the quality of observational studies and identify potential sources of bias. It consists of a set of questions that assess the risk of bias across six domains: study design, sample size, confounding, measurement of exposure and outcome, missing data, and analysis. Table 2 presents an assessment of the risk of bias for each included study. This table presents the results of the risk of bias assessment for each included study. The studies are divided into two categories: in vitro and in vivo studies. For each study, two reviewers independently rated the risk of bias across 11 domains or questions. The domains or questions assessed in the table are not specified, but they may include factors such as study design, sample size, confounding, measurement of exposure and outcome, missing data, and analysis. The risk of bias for each domain or question is rated as low (+), high (++), or unclear (-). The overall risk of bias for each study is determined based on the ratings across all domains or questions. It is important to consider the risk of bias when interpreting the results of a study, as biases can affect the validity and reliability of the findings. The main source of bias identified in these studies was the lack of blinding, which refers to whether research personnel were aware of the study groups and whether outcome assessments were blinded. Other potential sources of bias included the lack of information about statistical approaches, such as evaluating outliers and verifying normal distribution.

Table 3 presents information on the performance of flavonoid nanoparticles as a release system in vitro. The table includes data on the type and common name of the flavonoids, the composition of the nanoparticles in which they are encapsulated, the size and zeta potential of the nanoparticles, the encapsulation efficiency, and the release of flavonoids at 24 h. One of the key findings is that the performance of flavonoid nanoparticles as a release system can vary significantly depending on the type of flavonoid and the composition of the nanoparticles. For example, some nanoparticles have high encapsulation efficiency and are able to effectively enclose and protect the flavonoids inside the nanoparticle, while others have lower encapsulation efficiency. Additionally, the release of flavonoids from the nanoparticles over a period of 24 h can also vary significantly. In some cases, a high percentage of the flavonoids are released, while in other cases the release is lower. The table provides a comprehensive list of various types of flavonoids, including flavanones, flavones, flavonols, and flavanols. Among the commonly mentioned flavonoids in the table are naringenin, apigenin, chrysin, quercetin, hesperetin, kaempferol, fisetin, baicalein, and naringin.

The encapsulation strategy of naringenin, which is a flavanone found in citrus fruits, takes advantage of the structural features of naringenin, which features a 4,5,7-trihydroxyflavanone structure. Similarly, apigenin, a flavonoid present in plants like chamomile and parsley, possesses the 4’,5,7-trihydroxyflavone structure and is encapsulated in PLGA-PEG. Chrysin, another flavonoid found in the passionflower plant, is loaded into PLGA-PEG nanoparticles due to its 5,7-dihydroxyflavone structure. On the other hand, quercetin, a flavonol found in fruits and vegetables, including apples, berries, and onions, is incorporated into chitosan-modified monoolein nanoparticles. This choice of nanocarrier is based on the 2-(3,4-dihydroxyphenyl)-3,5,7-trihydroxy-4H-chromen-4-one structure of quercetin. Hesperetin, abundant in citrus fruits, is encapsulated in Hes Gd_2_(CO_3_)_3_@PDA nanoparticles, taking into account its 3’,5,7-trihydroxy-4-methoxyflavanone structure. Kaempferol, found in tea, broccoli, and kale, is loaded into hydroxyapatite nanoparticles, benefiting from its 3,5,7-trihydroxy-2-(4-hydroxyphenyl)-4H-1-benzopyran-4-one structure. Fisetin, present in strawberries, is encapsulated in an emulsion consisting of Miglyol^®^ 812N, Labrasol^®^, Tween^®^ 80, and Lipoid E80^®^, while baicalein, derived from Scutellaria baicalensis, is loaded into poly(ethylene glycol)-block-poly(D,L-lactide) nanoparticles. These choices of nanocarriers take into consideration the specific chemical structures of fisetin (2-(3,4-dihydroxyphenyl)-3,7-dihydroxychromen-4-one) and baicalein (5,6,7-trihydroxyflavone). Figure 3 presents the structures of the molecules and the skeleton of flavonoids.

Some of the key observations from the data in Table 3 highlight the relationship between nanoparticle size, encapsulation efficiency, and release behavior. Smaller nanoparticles, such as those made from Au-mPEG and chitosan with glucosamine hydrochloride, tend to exhibit higher encapsulation efficiency, ranging from 78% to 99%. These nanoparticles also display a wide range of release percentages at 24 h, varying from 35% to 90%. This suggests that their smaller size facilitates efficient encapsulation of the flavonoids, but the release behavior may be more variable. On the other hand, larger nanoparticles, like those made from PLGA and D-α-tocopherol PEG succinate-b-poly(β-thioester) copolymer, generally show lower encapsulation efficiency, ranging from 13.4% to 13%. These nanoparticles also exhibit relatively consistent release percentages at 24 h, close to 70%. This implies that their larger size may present challenges in achieving high encapsulation efficiency, but they may offer a more controlled and sustained release of the encapsulated flavonoids. Interestingly, some of the larger nanoparticles, such as those made from albumin and carboxymethyl dextran, L-cysteine, and octadecylamine, demonstrate lower encapsulation efficiency, ranging from 89% to 72.13%. However, these nanoparticles tend to have higher release percentages at 24 h, ranging from 60% to 90%. This suggests that while their encapsulation efficiency may be compromised, their larger size could facilitate a faster release of the encapsulated flavonoids. It is worth noting that the zeta potential of the nanoparticles plays a role in their stability and release behavior. A high zeta potential can contribute to improved stability of the nanoparticles in suspension, leading to higher encapsulation efficiency. However, it can also result in the reduced release of the flavonoids from the nanoparticles. Therefore, the specific relationship between zeta potential, encapsulation efficiency, and release behavior depends on the composition and conditions of the nanoparticles.

It should be emphasized that the relationship between zeta potential and the encapsulation/release of flavonoids is complex and influenced by several factors. One such factor is the composition of the nanoparticles. Different materials used in nanoparticle formulations can exhibit different surface charge properties, which can influence the zeta potential. Additionally, experimental conditions such as pH, ionic strength, and the presence of biological media can also affect the zeta potential and subsequently impact the release behavior of flavonoids. In the table, it can be observed that the zeta potential values vary among the different formulations. These variations suggest differences in the surface charge properties of the nanoparticles, which can affect their stability and release characteristics. It is important to consider that zeta potential alone is not the sole determinant of nanoparticle behavior. Other factors, such as particle size, composition, and surface modification, also contribute to the overall performance of flavonoid-loaded nanoparticles.

Larger nanoparticles tend to have lower surface area-to-volume ratios and may have lower zeta potentials. Also, the bigger the nanoparticle means that more material was required for the carrier, which may affect the loading capacity compared to smaller ones. This can lead to lower encapsulation efficiency but may also result in a higher release of the flavonoids from the nanoparticles. For example, some of the smaller nanoparticles in the table, such as those made from PLGA-PEG and chitosan-modified monoolein, have higher encapsulation efficiency, ranging from 56.6% to 99.4%. These nanoparticles also tend to have a lower release at 24 h, ranging from 15% to 30%.

As can be seen, polymeric nanoparticles predominate in both in vivo and in vitro studies. Recent studies have demonstrated the potential of nanoparticles in enhancing the bioavailability and therapeutic efficacy of various drugs. For instance, a systematic review and meta-analysis revealed that polymeric nanoparticulate systems can significantly augment the absorption and bioavailability of orally administered drugs compared to conventional formulations [60]. Similarly, a review article summarized the state-of-the-art optimization of formulations, including nanonization and encapsulation in nanoscale carriers, to improve the solubility, stability, and bioavailability of bioactive compounds, such as citrus flavonoids [61].

Some flavonoid nanoparticles have been found to have high encapsulation efficiency, meaning that they are able to effectively enclose and protect the flavonoid inside the nanoparticle. Table 4 presents data from in vivo studies in which the formulations were administered to animal models. The table includes information on the characteristics of the formulations. The data collected from experiments on cytokine production, either in vitro or in vivo, are also presented in Table 4. It mentions several materials that are commonly used in the fabrication of nanoparticles for the encapsulation and release of flavonoids. These materials include polymers, metals, and other types of biomaterials. Polymers are a common material used in the fabrication of nanoparticles due to their biocompatibility, biodegradability, and tunable properties. Some of the polymers mentioned in the table include poly(lactic-co-glycolic acid) (PLGA), polyethylene glycol (PEG), polyvinylpyrrolidone (PVP), and silk fibroin. These polymers can be used alone or in combination with other materials to create nanoparticles with specific properties.

When flavonoids are encapsulated in nanoparticles, their effects on cytokine production can be enhanced in both in vitro and in vivo settings. This enhancement can be attributed to several factors. Firstly, encapsulation in nanoparticles can improve the cellular uptake of flavonoids, thanks to the small size and surface characteristics of nanoparticles that facilitate their interaction with cell membranes. This increased cellular uptake results in higher intracellular concentrations of flavonoids, which can lead to more pronounced effects on cytokine production. Secondly, encapsulation provides protection to flavonoids, enhancing their stability and bioavailability. Flavonoids are known to have poor stability and low bioavailability, but encapsulation in nanoparticles can shield them from degradation and improve their absorption and circulation in the body. Additionally, nanoparticles can be designed to provide controlled and sustained release of encapsulated flavonoids, ensuring a gradual and prolonged delivery to target cells or tissues. This sustained release pattern may be more effective in modulating cytokine responses compared to the rapid release of non-encapsulated flavonoids. However, it is essential to acknowledge that the specific mechanisms underlying these changes may vary depending on the properties of the flavonoid, the type of nanoparticles used, and the experimental conditions employed.

The effect of cytokines in the inflammation cascade is to modulate the growth and activation of immune cells in response to pathogens and antigens; therefore, their downregulation allows for the reduction of the negative effects of the inflammatory response [62,63,64]. As an alternative, flavonoids have had nice activity; nevertheless, their bioavailability is limited, and this compromises their effect. However, the encapsulation of flavonoids with nanoparticles has been shown as an alternative to increase their bioavailability and thus their effectiveness in downmodulating some proinflammatory cytokines (Figure 3).

The interaction between the cell membrane and nanoparticles is another crucial aspect to consider when assessing the efficiency of flavonoid-loaded nanoparticles. Understanding the membrane–nanoparticle interaction is essential for evaluating the cellular uptake and bioavailability of flavonoids encapsulated in nanoparticles. For example, the establishment of hydrophobic–hydrophobic interactions between polyphenols and liposome membranes can enhance the encapsulation and delivery of polyphenols. The hydrophobic nature of polyphenols allows them to easily associate with the liposome membrane, resulting in the efficient loading of the polyphenols into the liposomes. This interaction can protect the polyphenols from degradation and enhance their stability during storage and transport. Additionally, the hydrophobic–hydrophobic interactions can facilitate the release of polyphenols from the liposomes at the target site, improving their bioavailability and therapeutic efficacy. The fisetin formulation encapsulated in liposomes with different compositions such as Chol/DOPC/DODA-PEG2000 and Isoscutellarein/egg phosphatidylcholine exhibits a size range of 130 to 298.3 nm, and encapsulation efficiencies ranging from 37.5% to 95% [65]. These values are consistent with the data reported for other nanoparticles included in our review. Liposomes have been extensively studied and widely used as drug-delivery systems due to their biocompatibility, versatility, and ability to encapsulate both hydrophilic and hydrophobic compounds. They can provide a controlled release of encapsulated polyphenols, protect them from degradation, and enhance their solubility.

The selected studies all relate to the use of flavonoid nanoparticles for the regulation of inflammation in various disease models. The selected articles conducted in vitro studies to evaluate the solubility of the encapsulation, the release of flavonoids through cells, and the effect of the encapsulated flavonoids on the expression or release of cytokines. They used a model of cell stimulation with lipopolysaccharide (LPS) to induce the expression and secretion of inflammatory cytokines in macrophages, fibroblasts, or other cell lines. The specifics of the model may vary depending on the cell line. After treating the cells with the encapsulated flavonoids and stimulating them with LPS for 24–72 h, the researchers harvested supernatants and RNAs for analysis of cytokine expression or quantification using PCR or Western blot. In the in vivo inflammation models, the researchers induced inflammation by administering intraperitoneal injections of LPS to the mice for a certain period. After a set period of LPS treatment, the mice were given orally administered encapsulated flavonoids for an additional period before being sacrificed to extract tissues for cytokine analysis using Western blot or PCR.

The encapsulation of flavonoids in nanoparticles has shown significant differences in the modulation of proinflammatory cytokines (Figure 4), mainly the group of interleukins and interferons, as shown in Table 4. In vitro assays Hesperetin in nanospongues [56], Kaempferol with hydroxyapatite [44] and in nanoparticles [58], quercetin with albumin [46] and in nanoparticles [42], as well as baicalein [54] and naringenin nanoparticles [55] were able to suppress IL-1β, IL-6, IL-8, and TNF-α levels with statistically significant difference between encapsulated and non-encapsulated flavonoid. On the other hand, the in vivo assays of Hesperetin nanoparticles alone inhibit TNFα levels [36]; in contrast, Wogonin nanoparticles [45], Eupafolin nanoparticles [39], Kaempferol nanocomplexes [49], and Quercetin nanoparticles [51] mainly decreased IL-1β, IL-6, IL-18, and TNF-α levels with significant differences between encapsulated flavonoids versus non-encapsulated flavonoids.

The main interleukins modulated by encapsulated flavonoids in both in vitro and in vivo studies are IL-1β, IL-6, and TNF-α. These cytokines are important regulators of inflammation, with IL-1β being produced by monocytes and macrophages and inducing the production of IL-6. TNF-α activates the regulation of other inflammatory mediators such as IL-1 and IL-6, and its activity at the cellular level attracts lymphocytes and neutrophils and promotes antigen recognition and tissue repair [66,67,68]. Some studies have found that encapsulated flavonoids can effectively downregulate these interleukins, while others have found no significant difference between encapsulated and free flavonoids. These findings suggest that the type of encapsulation and material used may be key in determining the effectiveness of flavonoids in downregulating interleukins.

The effects of encapsulated flavonoids on cytokine activity in cell culture models vary depending on the type of flavonoid and cell line used. In general, it seems that encapsulated flavonoids can downregulate the expressions of various cytokines such as IL-1β, IL-6, TNF-α, and IL-8. However, some studies have also found that encapsulated flavonoids can upregulate the expression of certain cytokines such as IL-10. The effects of encapsulated flavonoids on cytokine activity also appear to depend on the specific cell line used, with some cell lines showing greater effects than others. It is worth noting that the effects of encapsulated flavonoids may also depend on the concentration and duration of treatment.

The downregulation of IL-1β, IL-6, and TNF-α when using flavonoid-loaded nanoparticles can be attributed to their anti-inflammatory properties. Flavonoids have been shown to inhibit the production and release of these pro-inflammatory cytokines. They can modulate various signaling pathways involved in inflammation, such as NF-κB and MAPK pathways, thereby reducing the expression of pro-inflammatory cytokines. The upregulation of IL-2 and IL-10 when using flavonoid-loaded nanoparticles can be attributed to their immunomodulatory effects. Flavonoids have been shown to enhance the production and release of these anti-inflammatory cytokines. Flavonoids can stimulate the production of IL-2, thereby promoting the expansion and function of T-cells, which are important for immune surveillance and response. Flavonoids can induce the expression of IL-10, leading to the suppression of inflammation and the promotion of immune tolerance.

There may be some limitations to the review processes used in this systematic review. For example, the search strategy may not have been comprehensive enough to capture all relevant studies, and the inclusion criteria may have excluded some relevant studies. It is also possible that the quality of the included studies varied, which may have influenced the overall results. Despite these limitations, the results of this review provide valuable insights into the effects of encapsulated flavonoids on cytokine activity in cell culture models. In terms of implications for practice, policy, and future research, the findings of this review suggest that encapsulated flavonoids may be effective in downregulating certain cytokines, such as IL-1β, IL-6, and TNF-α, in cell culture models. This may have potential applications in the treatment of inflammation-related diseases. However, more research is needed to fully understand the mechanisms behind these effects and to determine the optimal conditions for using encapsulated flavonoids in clinical settings. Future research should also aim to replicate and extend the findings of this review in order to establish the generalizability and clinical relevance of these results.

## 4. Conclusions

In conclusion, it is important to acknowledge the limitations of our review process. For example, the search strategy may not have been comprehensive enough to capture all relevant studies, and the inclusion criteria may have excluded some relevant studies. Nevertheless, our review findings highlight the potential of encapsulated flavonoids in modulating cytokine activity in various in vitro and in vivo models. In vitro studies have demonstrated that encapsulated flavonoids can have an effect on cytokine activity in various cell culture models. The direction of this effect can vary, with some studies finding that encapsulated flavonoids can downregulate the expression of cytokines such as IL-1β, IL-6, TNF-α, and IL-8, while others have found that they can upregulate the expression of cytokines such as IL-10. The specific effect on cytokine activity appears to depend on the type of flavonoid and the cell line used, as well as the concentration and duration of treatment. Some studies have also found that the use of encapsulated flavonoids can have a greater effect on cytokine activity compared to non-encapsulated flavonoids. In the same way, several studies have shown that flavonoid encapsulation in nanoparticles can have a positive effect on cytokine regulation in different animal models, such as in the case of TNF-α, IL-1β, IL-6, and TGF-β1. However, in some cases, no significant difference was observed between the administration of encapsulated and non-encapsulated flavonoids, such as in the case of IL-10 and IL2. It is important to note that these findings are only a sample of the available studies, and more research is needed to fully understand how flavonoid encapsulation may affect cytokine activity in different cell lines, animal models and in human clinical conditions. Overall, this review highlights the potential of flavonoid nanoparticles as a release system for delivering flavonoids. Further research is needed to fully understand the mechanisms behind their performance and to optimize their design for different applications.

## Figures and Tables

**Figure 1 biomolecules-13-01158-f001:**
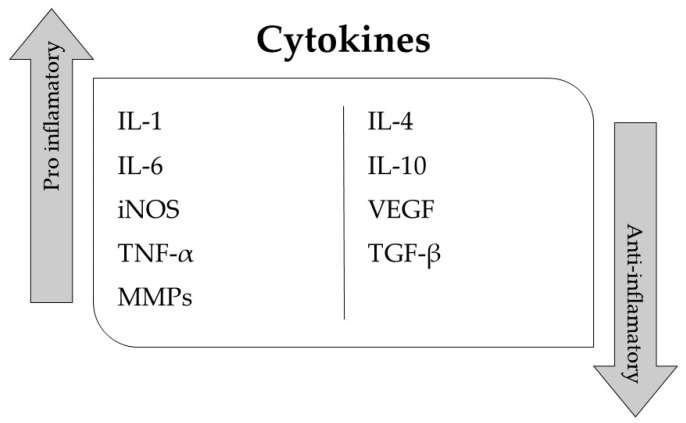
Classification of pro-inflammatory and anti-inflammatory cytokines based on their biological functions. The diagram illustrates the major cytokine families along with their respective pro-inflammatory or anti-inflammatory roles in immune responses.

**Figure 2 biomolecules-13-01158-f002:**
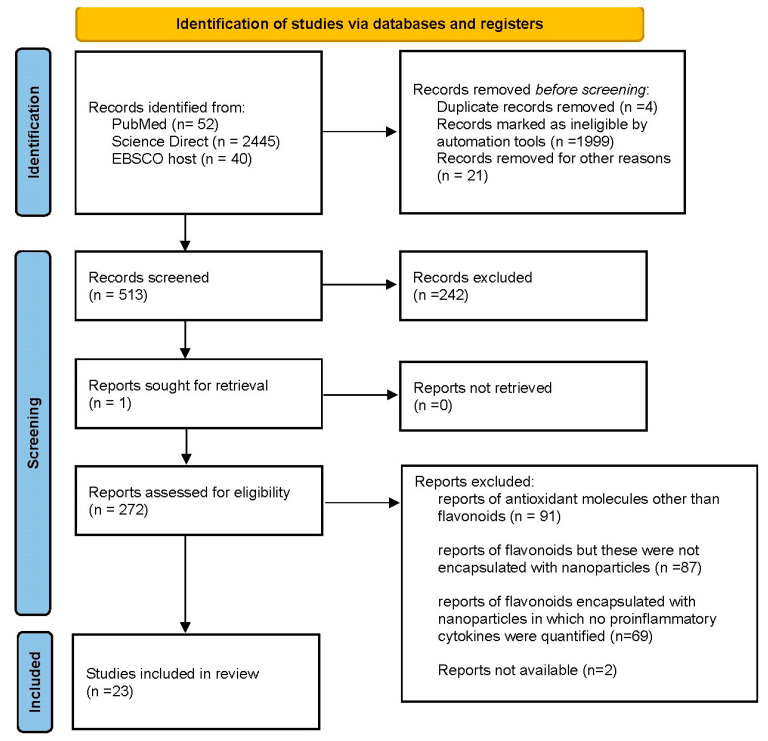
PRISMA diagram of literature searching and screening process.

**Figure 3 biomolecules-13-01158-f003:**
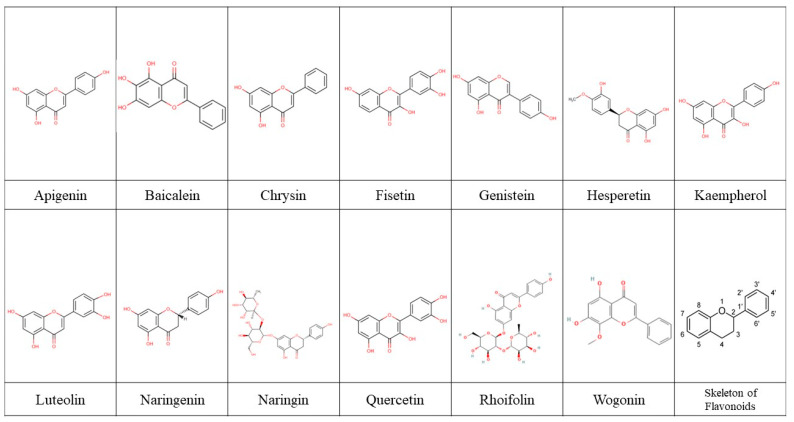
Skeleton of flavonoids. Structural skeleton of flavonoids, highlighting the common features including the carbon backbone, two aromatic rings, and a heterocyclic ring. Variations in the substitution patterns and functional groups contribute to the diverse range of flavonoid compounds.

**Figure 4 biomolecules-13-01158-f004:**
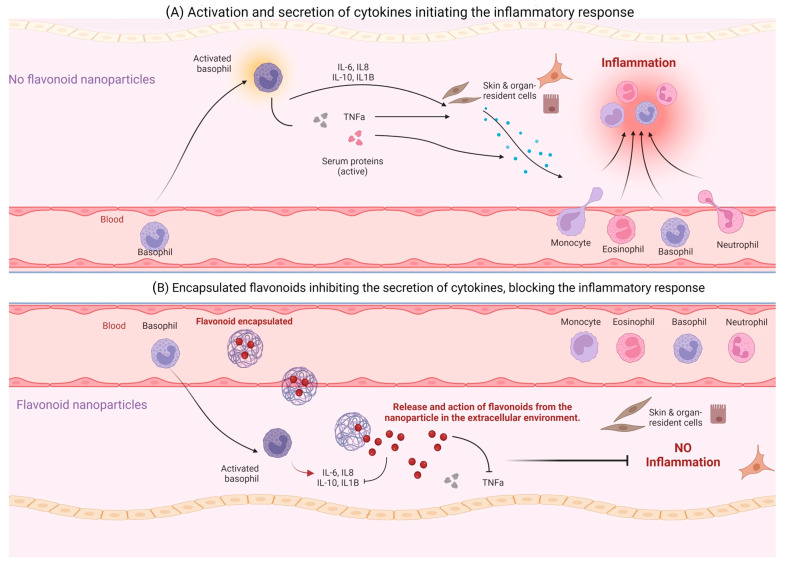
Schematic representation of the mechanism of action of flavonoid-encapsulated nanoparticles in modulating the inflammatory response by inhibiting cytokine action. The diagram illustrates how nanoparticles encapsulating flavonoids can effectively deliver these compounds to target sites, where they interact with cytokines involved in the inflammatory response, leading to the downregulation of pro-inflammatory cytokine production.

**Table 1 biomolecules-13-01158-t001:** Main characteristics of the selected studies.

Reference ^1^	Objective/Study Description
Krishnan et al., 2017 [36]	The objective of this in vivo study was to investigate the protective effect of Au-mPEG(5000)-S-HP nanoparticles on the expressions of cell inflammation and cell proliferation during diethylnitrosamine-induced hepatocarcinogenesis in male Wistar albino rats. The specific aims of the study were to determine the role of the Au-mPEG(5000)-S-HP nanoparticles in modulating inflammation and proliferation in the liver, and to assess the potential protective effects of these nanoparticles against diethylnitrosamine-induced hepatocarcinogenesis.
Kumar R. and Abraham, 2017 [37]	The objective of this in vitro study was to evaluate the inhibitory effects of novel polyvinyl pyrrolidone-coated naringenin nanoparticles (NAR NPs) on lipopolysaccharide (LPS)-induced inflammation in RAW 264.7 mouse macrophages. The specific aims of the study were to determine the ability of the NAR NPs to inhibit inflammation in response to LPS stimulation, and to investigate the mechanisms of action of the NAR NPs in modulating inflammation in macrophages.
J. Zhang et al., 2017 [38]	The objective of this in vitro study was to evaluate the impact of apigenin on fibroblast functions in the context of pulmonary inflammation and fibrosis. The specific aims of the study were to investigate the effects of apigenin on fibroblast proliferation, cell cycle arrest, and the production of profibrosis and proinflammation cytokines in bronchoalveolar lavage (BAL) and primary pulmonary cells.
H. Zhang et al., 2017 [39]	The objective of this in vitro study was to evaluate the effect of eupafolin release through nanoparticles on oxidative stress-related acute kidney injury and inflammatory response in animal models. The specific aims of the study were to determine whether eupafolin-encapsulated nanoparticles could ameliorate oxidative stress and inflammation in the kidney, and to investigate the potential mechanisms of action through the NF-kB and MAPK signaling pathways.
Firouzi-Amandi et al., 2018 [40]	The objective of this in vitro study is to investigate the potential of chrysin-loaded PLGA-PEG nanoparticles (NPs) to modulate the functional polarity of macrophages towards anti-inflammatory M2 phenotypes. Chrysin is a natural compound that has been shown to have immunomodulatory properties. PLGA-PEG is a type of biodegradable, biocompatible polymer that is often used in drug delivery due to its non-immunogenic and non-toxic properties.
vanden Braber et al., 2018 [41]	The objective of this in vivo study is to investigate the potential of microencapsulated soy genistein (Ge) to preserve its biological activity and attenuate clinical signs of acute colitis in mice. The researchers in this study aimed to use water-soluble chitosan obtained by Maillard reaction with glucosamine to microencapsulate Ge and preserve its biological activity for oral administration. They also investigated the release of Ge, which was found to be pH-dependent with a super Case II mechanism at pH 1.2 and anomalous transport with non-Fickian kinetics at pH 6.8.
Cherk Yong et al., 2019 [42]	The objective of this in vitro study was to evaluate the effects of quercetin-loaded large unilamellar vesicles (LCNs) and small unilamellar vesicles (sm-LCNs) on the secretion of proinflammatory cytokines in an immortalized human bronchial epithelial cell line (BCi-NS1.1) induced with lipopolysaccharide (LPS). The specific aims of the study were to investigate the ability of the quercetin-loaded LCNs and sm-LCNs to modulate cytokine production in response to LPS stimulation, and to determine the mechanisms of action of these formulations in modulating inflammation.
Ouyang et al., 2019 [43]	The objective of this study in vitro was to investigate the mechanisms of action of a nanomedical therapy called Gd_2_(CO_3_)_3_@PDA(Hes)-PEG-DWpeptide in the treatment of osteoarthritis (OA). This therapy is designed to protect cartilage and may have potential as a treatment for OA. The study aims to clearly illustrate the mechanisms of action of this therapy to understand how it works and potentially improve its effectiveness in treating OA.
Pang et al., 2019 [44]	The objective of this in vitro study was to evaluate the effects of hydroxyapatite (HAP) particles with different morphologies and sizes, as well as their biomolecule-loaded particles, on the inflammatory regulation of macrophage-like cells. This study aims to investigate how HAP particles with different morphologies and sizes, and biomolecule-loaded particles, may affect the inflammatory regulation of macrophage-like cells to understand their potential as a therapeutic intervention for inflammatory diseases.
Bei et al., 2020 [45]	The objective of this in vivo study was to evaluate the use of Wog NP to attenuate the cardio-protective role of Wog np against iso-induced myocardial infarction (MI). The specific aims of the study were to determine the ability of Wog np to protect against MI induced by iso and to investigate the mechanisms of action of Wog np in modulating cardiac injury and inflammation. The study aimed to assess the potential therapeutic benefits of Wog NP for the treatment of MI and other cardiovascular conditions.
Saha et al., 2020 [46]	The objective of this in vitro study was to evaluate the anti-inflammatory activity of quercetin nanoparticles in bone marrow-derived dendritic cells against crystal-induced inflammation. Dendritic cells are a type of immune cell that play a crucial role in the immune response to infection and inflammation. The study aims to investigate whether quercetin nanoparticles can reduce inflammation in dendritic cells when they are exposed to crystals that are known to cause inflammation. The results of this study could potentially be used to develop a therapy using quercetin nanoparticles to treat inflammation.
Wang et al., 2020 [47]	The objective of this in vivo study is to improve the bioavailability and inflammation suppression potential of NAR by developing a novel self-nanomicellizing formulation containing NAR (RA-NAR). NAR is a compound that has been shown to have anti-inflammatory properties. However, its bioavailability, or the amount of the compound that is absorbed and becomes active in the body, is relatively low.
Xu et al., 2020 [48]	The objective of this in vivo study is to investigate whether fisetin nanoparticles (FN) can inhibit PM2.5-induced metabolic disorder and neuroinflammation by regulating astrocyte-activation-related NF-κB signaling. Fisetin is a flavonoid that has been shown to have anti-inflammatory and neuroprotective properties. PM2.5 is a type of air pollution particle that has been linked to various health problems, including metabolic disorders and neuroinflammation. The researchers in this study aimed to investigate whether fisetin nanoparticles could inhibit the negative effects of PM2.5 on metabolism and the brain by regulating astrocyte activation-related NF-κB signaling.
F. Zhang et al., 2020 [49]	The objective of this study in vitro is to compare the pharmacokinetics, bioavailability, and brain distribution of a nasal micellar formulation of BE with the unformulated BE formulation administered orally in mice. BE is a compound that has been shown to have anti-inflammatory properties. The study aims to investigate whether the nasal micellar formulation of BE, which is a type of drug-delivery system, is more effective at delivering BE to the body and brain than the unformulated BE administered orally.
L. Zhang et al., 2020 [50]	The objective of this study in vitro was to validate the concept of using PVP as the only excipient to solubilize KAE to form a stable nanoaqueous ophthalmic formulation. PVP is a polymer that can be used to solubilize, or dissolve, other compounds in aqueous solutions. KAE is a compound that has anti-inflammatory and antioxidant properties. The formulation would be administered topically to the eye.
Diez-Echave et al., 2021 [51]	The objective of this in vivo study is to evaluate the intestinal anti-inflammatory effects of quercetin when it is administered in silk fibroin nanoparticles in a mouse model of colitis. The study aims to investigate whether quercetin administered in silk fibroin nanoparticles can reduce inflammation in the intestine of mice with colitis.
Al-Shalabi et al., 2022 [52]	The objective of this in vivo study is to evaluate the antioxidant and anti-inflammatory activities of optimized nanoparticles with rhoifolin in vitro in murine macrophages and in vivo in a rat paw edema model. The researchers in this study aimed to optimize the synthesis and properties of nanoparticles and evaluate their antioxidant and anti-inflammatory activities. They likely synthesized the nanoparticles and tested their effects on murine macrophages in vitro, as well as in vivo in a rat paw edema model, which is a commonly used model to study inflammation.
He et al., 2022 [53]	The objective of this study in vitro was to evaluate the ability of ADEX wall material to establish a close connection between cells and QNMs, which may favor the uptake of QCT by macrophages and impact the antioxidant and inflammatory capacity. QNMs (quantum dot nanomaterials) are a type of nanomaterial that have unique optical and electronic properties due to their size and shape. ADEX wall material is a type of material that is used to coat surfaces and is known for its ability to interact with cells. The researchers in this study sought to investigate whether the interaction between ADEX wall material and QNMs could favor the uptake of QCT (quantum dot conjugates) by macrophages and impact their antioxidant and inflammatory capacity.
Li et al., 2022 [54]	The objective of this in vitro study is to evaluate the inflammatory modulation effects of BE and a BE-adsorbed nanosystem in oral cells by targeting the canonical NF-kB pathway. BE is a compound that has been shown to have anti-inflammatory properties. The study aims to investigate whether BE and a BE-adsorbed nanosystem can modulate inflammation in oral cells by targeting the canonical NF-kB pathway, which is a signaling pathway involved in the regulation of inflammation.
Mohamed et al., 2022 [55]	The objective of this in vitro study is to investigate the ability of a naringin-dextrin nanoformula (NDN) to enhance the therapeutic action of naringin against human liver cancer. Naringin is a flavonoid that has been shown to have anti-cancer properties. The researchers in this study sought to develop a nanoformula containing naringin and dextrin in order to improve the bioavailability and therapeutic action of naringin.
Rodrigues et al., 2022 [56]	The objective of this study is to design ethyl cellulose porous nanosponges as a colloidal carrier for the topical delivery of hesperetin. Hesperetin is a flavonoid that has been shown to have various health benefits, including antioxidant and anti-inflammatory effects. Ethyl cellulose is a type of polymer that is often used in drug delivery due to its biocompatibility and ability to form porous structures. The researchers in this study aimed to design ethyl cellulose porous nanosponges as a colloidal carrier for the topical delivery of hesperetin. The study likely involved synthesizing the ethyl cellulose nanosponges and incorporating hesperetin into them, and then testing their properties and effectiveness in in vitro models.
Tan et al., 2022 [57]	The objective of this in vivo study is to evaluate the anti-UC efficacy of LUT@TPGS-PBTE nanoparticles in a mouse model of colitis and explore the molecular mechanism and modulation of the pathological microenvironment of this system for UC treatment. LUT (luteolin) is a flavonoid that has been shown to have various health benefits, including antioxidant and anti-inflammatory effects. TPGS (d-alpha-tocopheryl polyethylene glycol 1000 succinate) is a type of molecule that can enhance the solubility and bioavailability of drugs. PBTE (polybutylene terephthalate) is a type of polymer that is often used in drug delivery due to its biocompatibility and ability to form nanoparticles. The researchers in this study aimed to synthesize LUT@TPGS-PBTE nanoparticles and evaluate their anti-UC efficacy in a mouse model of colitis. UC (ulcerative colitis) is a type of inflammatory bowel disease (IBD) that affects the colon.
Yang et al., 2022 [58]	The objective of this in vitro study is to characterize the properties of kaempferol-loaded fibroin nanoparticles and evaluate their bioreactive release, cytotoxicity, cell internalization, ROS scavenging activity, and anti-inflammatory ability. Kaempferol is a flavonoid that has been shown to have various health benefits, including antioxidant and anti-inflammatory effects. Fibroin is a protein that is found in silk and has been used as a material for drug delivery due to its biocompatibility and biodegradability. The researchers in this study aimed to synthesize kaempferol-loaded fibroin nanoparticles and investigate their properties, including their bioreactive release, cytotoxicity, cell internalization, ROS scavenging activity, and anti-inflammatory ability.

^1^ References are listed in chronological order by year of publication. In the other tables, studies are separated based on in vitro or in vivo.

**Table 2 biomolecules-13-01158-t002:** Risk of bias assessment for included studies using the OHAT Rob tool.

Reference		Risk-of-Bias Questions or Domains *										
		1	2	3	4	5	6	7	8	9	10	11
In vitro studies												
Kumar R. and Abraham, 2017 [37]	Reviewer 1	++	++			+	+	+	++	+	++	-
	Reviewer 2	++	+			+	-	+	++	++	++	+
J. Zhang et al., 2017 [38]	Reviewer 1	+	-			+	+	++	++	+	++	--
	Reviewer 2	++	+			+	+	++	++	+	+	--
Firouzi-Amandi et al., 2018 [40]	Reviewer 1	-	-			+	+	+	+	-	+	--
	Reviewer 2	+	+			+	+	+	-	+	-	--
Cherk Yong et al., 2019 [42]	Reviewer 1	-	-			-	-	+	+	-	+	--
	Reviewer 2	-	-			-	-	+	+	-	-	--
Ouyang et al., 2019 [43]	Reviewer 1	+	+			+	+	++	+	+	+	-
	Reviewer 2	+	+			+	++	+	+	+	+	-
Pang et al., 2019 [44]	Reviewer 1	+	+			+	++	+	+	+	+	+
	Reviewer 2	+	+			+	+	+	+	+	+	-
Saha et al., 2020 [46]	Reviewer 1	+	+			+	+	+	+	+	+	+
	Reviewer 2	+	+			+	+	+	+	-	++	+
Xu et al., 2020 [48]	Reviewer 1	++	+			+	++	+	+	+	+	+
	Reviewer 2	+	+			+	++	+	+	++	+	+
L. Zhang et al., 2020 [50]	Reviewer 1	+	++			+	+	++	++	+	+	++
	Reviewer 2	+	++			+	+	+	+	+	+	++
He et al., 2022 [53]	Reviewer 1	+	-			+	+	+	+	+	-	--
	Reviewer 2	+	+			+	+	+	+	+	-	--
Li et al., 2022 [54]	Reviewer 1	+	+			+	+	+	+	+	+	-
	Reviewer 2	+	+			+	+	++	+	+	+	-
Mohamed et al., 2022 [55]	Reviewer 1	+	++			++	+	+	+	+	+	+
	Reviewer 2	+	+			++	+	+	+	+	+	+
Rodrigues et al., 2022 [56]	Reviewer 1	+	-			+	-	-	-	-	+	--
	Reviewer 2	+	+			+	+	+	-	-	-	--
Yang et al., 2022 [58]	Reviewer 1	+	+			+	+	+	+	+	+	+
	Reviewer 2	+	+			++	+	+	+	+	+	+
In vivo studies												
Krishnan et al., 2017 [36]	Reviewer 1	+	+	+	+	+	+	+	+	+	+	+
	Reviewer 2	+	+	-	+	+	+	+	+	+	+	+
H. Zhang et al., 2017 [39]	Reviewer 1	+	+	+	+	+	+	+	+	+	+	+
	Reviewer 2	-	-	+	+	+	+	+	+	+	+	+
vanden Braber et al., 2018 [41]	Reviewer 1	-	+	+	-	-	+	+	+	+	+	+
	Reviewer 2	-	+	+	-	-	+	+	+	+	+	+
Bei et al., 2020 [45]	Reviewer 1	+	+	+	+	+	+	+	+	+	+	+
	Reviewer 2	+	+	+	+	+	+	+	+	+	+	+
Wang et al., 2020 [47]	Reviewer 1	++	++	+	++	+	++	+	+	+	+	+
	Reviewer 2	+	++	++	+	+	+	++	+	+	+	+
F. Zhang et al., 2020 [49]	Reviewer 1	+	+	+	+	+	+	+	+	+	+	+
	Reviewer 2	+	+	+	+	+	+	+	+	+	+	+
Diez-Echave et al., 2021 [51]	Reviewer 1	+	+	+	+	+	+	+	+	+	+	+
	Reviewer 2	+	+	+	+	+	+	+	+	+	+	+
Al-Shalabi et al., 2022 [52]	Reviewer 1	+	++	++	++	+	+	+	+	++	+	+
	Reviewer 2	+	+	++	+	+	+	+	+	++	+	+
Tan et al., 2022 [57]	Reviewer 1	+	+	+	-	+	+	+	+	+	+	+
	Reviewer 2	+	+	-	-	+	+	+	-	+	+	+

* Potential sources of bias are assessed with a set of 11 questions. 1 Was administered dose or exposure level adequately randomized? 2 Was allocation to study groups adequately concealed? 3 Did selection of study participants result in the appropriate comparison groups? 4 Did study design or analysis account for important confounding and modifying variables? 5 Were experimental conditions identical across study groups? 6 Were research personnel blinded to the study group during the study? 7 Were outcome data complete without attrition or exclusion from analysis? 8 Can we be confident in the exposure characterization? 9 Can we be confident in the outcome assessment (including blinding of assessors)? 10 Were all measured outcomes reported? 11 Were there no other potential threats to internal validity.

**Table 3 biomolecules-13-01158-t003:** Characteristics of nanoparticles as flavonoid delivery systems.

Reference	Flavonoid	Nanoparticles in Whose Flavonoids are Encapsulated ^1^		Performance as a Flavonoid Release System	
	Common Name Type IUPAC Name	Composition	Size, nm Zeta Potential, mV	Encapsulation Efficiency, %	Release at 24 h, %
In vitro studies					
Kumar R. and Abraham, 2017 [37]	Naringenin Flavanone 4, 5, 7-trihydroxyflavanone	Hybrid nanoparticles	111 - ^2^	99.93	-
J. Zhang et al., 2017 [38]	Apigenin Flavone 4′,5,7-trihydroxyflavone	Polymeric nanoparticles PLGA-PEG	163 −18.6	56.6	60
Firouzi-Amandi et al., 2018 [40]	Chrysin Flavone 5,7-di-OH-flavone	Polymeric nanoparticles PLGA-PEG	205 −6.3	88	40
Cherk Yong et al., 2019 [42]	Quercetin Flavonol 2-(3,4-dihydroxyphenyl)-3,5,7-trihydroxy-4H-chromen-4-one	Polymeric nanoparticles Chitosan-modified monoolein	223.9 −15.6	99.4	30
Ouyang et al., 2019 [43]	Hesperetin Flavanone 3′, 5,7-trihydroxy-4-methoxyflavanone	Core-shell nanoparticles Hes Gd_2_(CO_3_)_3_@PDA nanoparticles	70 −7.5	67.86	70
Pang et al., 2019 [44]	Kaempferol Flavonol 3,5,7-trihidroxi-2-(4-hidroxifenyl)-4H-1-benzopiran-4-ona	Inorganic nanoparticles Hydroxyapatite	100 −10.85	90	15
Saha et al., 2020 [46]	Quercetin Flavonol 2-(3,4-dihydroxyphenyl)-3,5,7-trihydroxy-4H-chromen-4-one	Protein-based nanoparticles Albumin	322 -	89	60
Xu et al., 2020 [48]	Fisetin Flavonol 2-(3,4-dihydroxyphenyl)-3,7-dihydroxychromen-4-one	Nanoemulsion Emulsion with Miglyol^®^ 812N/Labrasol^®^/Tween^®^ 80/Lipoid E80^®^	153 ^3^ −28.4	-	-
L. Zhang et al., 2020 [50]	Baicalein Flavone 5,6,7-trihydroxyflavone	Polymeric nanoparticles Poly (ethylene glycol)-block-poly (D, L-lactide)	24.98 −7.33	69.85	20
He et al., 2022 [53]	Quercetin Flavonol 2-(3,4-dihydroxyphenyl)-3,5,7-trihydroxy-4H-chromen-4-one	Complex coacervation-based nanoparticle system Carboxymethyl dextran, L-cysteine, and octadecylamine	372 31.4	72.13	60 ^4^
Li et al., 2022 [54]	Baicalein Flavone 5,6,7-trihydroxyflavone	Core-shell nanoparticles Silica capped with poli disulfide	88.4 22.8	5.2	18
Mohamed et al., 2022 [55]	Naringin Flavanone (2S)-4′,5-Dihydroxy-7-[α-L-rhamnopyranosyl-(1→2)-β-D-glucopyranosyloxy]flavan-4-one	Polymeric nanoparticles Dextrin	77.4 −31.6	35	90 ^5^
Rodrigues et al., 2022 [56]	Hesperetin Flavanone 3′, 5,7-trihydroxy-4-methoxyflavanone	Polymeric nanoparticles Ethylcellulose—PVA	105.08 −1.35	-	40
Yang et al., 2022 [58]	Kaempferol Flavonol 3,5,7-trihidroxi-2-(4-hidroxifenyl)-4H-1-benzopiran-4-ona	Protein-based nanoparticles Fibroin	151 −25.2	53.8	15
In vivo studies					
Krishnan et al., 2017 [36]	Hesperetin Flavanone 5, 7, 3′-trihydroxy-4′methoxy	Core-shell nanoparticles Au-mPEG (5000)	120 −8.42	99	90
H. Zhang et al., 2017 [39]	Eupafolin	Polymeric nanoparticles Eudragit E100—PVA	90.8-	-	-
vanden Braber et al., 2018 [41]	Genistein Isoflavone 4′,5,7-Trihydroxyisoflavone	Biopolymeric nanoparticles Chitosan with glucosamine hydrochloride	2.62 -	78	35
Bei et al., 2020 [45]	Wogonin Flavone 5,7-dihydroxy-8-methoxy flavone	Polymeric nanoparticles PLGA	194.8 −37.09	74.89	73.28
Wang et al., 2020 [47]	Naringenin Flavanone (2S)-4′,5,7-Trihydroxyflavan-4-one	Natural product-based nanoparticles Rebaudioside A (steviol glycoside)	5.234 −2.268	-	-
F. Zhang et al., 2020 [49]	Kaempferol Flavonol 3,5,7-trihidroxi-2-(4-hidroxifenyl)-4H-1-benzopiran-4-ona	Polymeric nanoparticles Polyvinylpyrrolidone (PVP)	8.62 −5.31	-	-
Diez-Echave et al., 2021 [51]	Quercetin Flavonol 2-(3,4-dihydroxyphenyl)-3,5,7-trihydroxy-4H-chromen-4-one	Protein-based nanoparticles Silk fibroin	175.8 −24.5	18.12	-
Al-Shalabi et al., 2022 [52]	Rhoifolin Flavone 7-[4,5-dihydroxy-6-(hydroxymethyl)-3-(3,4,5-trihydroxy-6-methyloxan-2-yl)oxyoxan-2-yl]oxy-5-hydroxy-2-(4-hydroxyphenyl)chromen-4-one	Polymeric nanoparticles PLGA followed by tannic acid-mediated surface modification with PEG	204 −28	45	80
Tan et al., 2022 [57]	Luteolin Flavone 3′,4′,5,7-Tetrahydroxyflavone	Polymeric nanoparticles D-α-tocopherol PEG succinate-b-poly(β-thioester) copolymer	426.3 −18	13.4	85

^1^ The physicochemical properties correspond to those nanoparticles that already include the encapsulated flavonoids. ^2^ Unavailable data are represented by the sign (-). ^3^ The article indicated that “For fisetin nanoparticles, the nanoemulsion particles were prepared according to Ragelle et al. (2012) [59]”; therefore, the data were taken from the indicated reference. ^4^ The study reported 9 h as the duration of the release experiment. ^5^ The study reported 12 h as the duration of the release experiment.

**Table 4 biomolecules-13-01158-t004:** The role of encapsulated flavonoids in modulating cytokine activity.

Reference	Cell Culture/Murine Model	Effect of Encapsulated Flavonoid on Cytokine Activity	
In vitro			
Kumar R. and Abraham, 2017 [37]	Macrophage cell line RAW264.7	Downregulation IL-1β IL-6 TNF-α	Naringenin nanoparticles downregulated the expressions of IL -6, IL-1β, and TNF–α, but no difference between encapsulated flavonoid vs. non-encapsulated flavonoid
J. Zhang et al., 2017 [38]	Primary pulmonary fibroblasts from Sprague Dawley rats	Downregulation IL-8	Apigenin nanoparticles had no effect on IL-6, selectively regulated the expression of IL-8, but no difference between encapsulated flavonoid vs. non-encapsulated flavonoid
Firouzi-Amandi et al., 2018 [40]	Primary peritoneal macrophages from C57BL/6 mouse	Downregulation IL-1β IL 6 TNF-α	Chyrsin nanoparticles effectively suppressed the expression levels of TNF-α, IL-1β, and IL-6 with a difference between encapsulated flavonoid vs. encapsulated only
Cherk Yong et al., 2019 [42]	Bronchial epithelial cell line BCi-NS1.1	Downregulation IL-1β IL-6 IL-8	Quercetin nanoparticles decrease the concentration of IL-1β, IL-6 and IL-8, with a difference between encapsulated flavonoid vs. non-encapsulated flavonoid
Ouyang et al., 2019 [43]	Primary chondrocytes from C57BL/6/ Bkl mice	Downregulation IL-1β	Hesperetin nanoparticles inhibited IL-1β, but no difference between encapsulated flavonoid vs. encapsulated only
Pang et al., 2019 [44]	Monocyte cell line THP-1, differentiated into macrophage cells	Downregulation IL-1β IL-6 TNF-α Upregulation IL-2 IL-10	Kaempferol-HAP only downregulated IL-1β, IL-6, TNF-α with a significant difference between encapsulated Flavonoid vs. non-encapsulated and flavonoid alone
Saha et al., 2020 [46]	Primary dendritic CD11c positive cells from femurs of C57BL/6 mice	Downregulation IL-1β IL-6 TNFα	Quercetin nanoparticles caused a decrease of IL-1 𝛽, IL-6, and TNF-𝛼, but only IL-6 has a significant difference between encapsulated flavonoid vs. non-encapsulated
Xu et al., 2020 [48]	Primary astrocytes from C57BL/6 mice	Downregulation IL1β IL-6 IL-8 TNF-α	Fisetin nanoemulsion 10 µg/L treatment has the ability to suppress IL-1β, IL-6, TNF-α, and IL-8 mRNA levels in PM2.5-induced cortex, a significant difference between encapsulated flavonoid vs. control, untreated cell
L. Zhang et al., 2020 [50]	Mouse microglia cell line BV-2	Downregulation IL-6 TNF-α	Baicalein nanoparticle decreased only IL-6 with a significant difference between encapsulated flavonoid vs. control, untreated cell
He et al., 2022 [53]	Murine macrophage cell line RAW264.7	Upregulate IL-10	Quercetin nanomicelles upregulate IL-10 in the LPS-induced cell inflammation model in vitro with better performances than free quercetin, a significant difference between encapsulated flavonoid vs. control, untreated cell
Li et al., 2022 [54]	Human gingival epithelial cell line hGECs	Downregulate IL-6 IL-8	Baicalein nanoparticles on decreased levels of IL-6 and IL-8 had a significant difference between encapsulated flavonoid vs. non-encapsulated flavonoid
Mohamed et al., 2022 [55]	Human liver cancer cell line HepG2	Downregulate IL-8.	Naringin nanoparticles on IL-8 had significant difference between encapsulated flavonoid vs. non-encapsulated flavonoid
Rodrigues et al., 2022 [56]	Macrophage cell line RAW264.7	Downregulation IL-1β IL6	Hesperetin in nanospongues inhibit the expression of I IL-6 and IL-1β, significantly different between encapsulated flavonoid vs. non-encapsulated flavonoid
Yang et al., 2022 [58]	Macrophage cell line RAW264.7	Downregulate TNFα	Kaempherol nanoparticles were dose-dependent on TNF-α, with a significant difference between encapsulated flavonoid vs. non-encapsulated flavonoid
In vivo			
Krishnan et al., 2017 [36]	Wistar strain albino rats, liver cancer model Dose: 20 mg/kg Administration route: intraperitoneal	Downregulation TNF-α	Hesperetin nanoparticles significantly inhibit levels of TNFα with significant difference between encapsulated flavonoid vs. non-encapsulated flavonoid
H. Zhang et al., 2017 [39]	C57BL/6 mice, inflammation LPS model, kidney Dose: 10 and 20 mg/kg Administration route: oral	Downregulation IL-1β IL6 IL-18	No effect on IL-10, but in IL1β, IL-6, IL-18 was significantly different between encapsulated flavonoid vs. non-encapsulated flavonoid
vanden Braber et al., 2018 [41]	C57BL/6 (B6) mice, inflammation model, luminal portion from colon Dose: 2.30 ± 0.20% (*w*/*w*) Administration route: Oral	Downregulation IL-10	Genistein nanoparticles have no significant differences between encapsulated flavonoid vs. encapsulated only
Bei et al., 2020 [45]	Adult male albino Wistar rats, heart tissue Dose: 25 and 50 mg/kg Administration route: oral	Downregulation IL-1 β IL-6 TNF-α	Wogonin nanoparticles in rodents decreased IL-1β, IL-6, and TNF-α significantly different between encapsulated flavonoid vs. non-encapsulated flavonoid
Wang et al., 2020 [47]	Adult male Sprague Dawley rats, small intestine injuries model Dose: 90 mg/kg Administration route: oral	Downregulation IL-1β IL-6	Naringenin nanoparticles reduce IL-1β and IL-6, with no significant difference between encapsulated flavonoid and the unencapsulated flavonoid
F. Zhang et al., 2020 [49]	Healthy New Zealand white rabbits Dose: 5 μL of a solution containing 4.5 mg/mL of KAE, 90 mg/mL of 17PF and a sodium hyaluronate ophthalmic solution (1 mg/mL) six times daily Administration route: Corneal	Downregulation IL-6 TGF-β1	Kaempferol nanocomplexes reduce IL-6 and TGF-β1 levels similar to free kaempferol, with significant differences between Encapsulated flavonoid vs. non-encapsulated flavonoid
Diez-Echave et al., 2021 [51]	Male C57BL/6J mice, model of mouse colitis Dose: 5 mg of quercetin/kg in 200 μL of PBS solution Administration route: Oral	Downregulation Il-1β IL-6 TNF-α	Quercetin nanoparticles decrease TNF-α, Il-1β, and IL-6 levels with significantly different between encapsulated flavonoid vs. non-encapsulated flavonoid
Al-Shalabi et al., 2022 [52]	Male Wistar rat, paw edema model Dose: 10 mg kg^−1^ Administration route: Intraperitoneal	Downregulation TNF-α IL-1β	Rhoifolin in film decreases IL-1β and TNF-α, with no significant difference between encapsulated flavonoid vs. non-encapsulated flavonoid
Tan et al., 2022 [57]	Male Kunming mice, model of mouse colitis Dose: 2 mg/kg Administration route: Oral	Downregulation IL2 IL6 IL17 TNFα Upregulation IL-10	Luteolin nanoparticles significantly upregulated IL-10 between encapsulated and non-encapsulated flavonoids, but in the inhibition of IL2, IL6, IL17 and TNFα there was no significant difference between encapsulated and non-encapsulated flavonoid.

## Data Availability

The data presented in this study are openly available in Zenodo at https://doi.org/10.5281/zenodo.7478273 (accessed on 1 July 2023).

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
