# Peer review of "Nanoparticle-Mediated Delivery of Flavonoids: Impact on Proinflammatory Cytokine Production: A Systematic Review"

_biomolecules, 2023, doi:10.3390/biom13071158_

Round 1

Reviewer 1 Report

biomolecules-2501441

Nanoparticle-Mediated Delivery of Flavonoids: Impact on Pro-2 inflammatory Cytokine Production. A Systematic Review

The manuscript by Barrón et al. summarized different nanosystems for the delivery of flavonoids with a focus on the pro-inflammatory cytokine production effects. The authors followed PRISMA guidelines to conduct a systematic review. They identified 23 eligible studies to incorporate into the review. Overall, the manuscript was well prepared and comprehensive. This is a revised version of a previous manuscript. Although it was improved, some previous concerns were not resolved. Below are specific comments to improve it.

1. Search strategy: as mentioned in the previous review, searching with “(Flavonoid) AND (nanoparticles) AND (cytokines) AND (in vitro/ in vivo)” can result in the omission of some papers. For example, flavonoids include a group of various compounds. Search with “flavonoid” may result in missing some studies where the compound names were mentioned instead of the term “flavonoid”. How did the author manage to avoid this?

2. Table 3 and relevant parts: it is more informative if the authors mention the nanosystem type (such as polymeric nanoparticles, micelles, lipid nanoparticles, etc.).

3. In Table 3, the last column, “%” should be removed from the data as it was mentioned in the column header.

Author Response

Reviewer 1

Nanoparticle-Mediated Delivery of Flavonoids: Impact on Proinflammatory Cytokine Production. A Systematic Review

The manuscript by Barrón et al. summarized different nanosystems for the delivery of flavonoids with a focus on the pro-inflammatory cytokine production effects. The authors followed PRISMA guidelines to conduct a systematic review. They identified 23 eligible studies to incorporate into the review. Overall, the manuscript was well prepared and comprehensive. This is a revised version of a previous manuscript. Although it was improved, some previous concerns were not resolved. Below are specific comments to improve it.

  1. Search strategy: as mentioned in the previous review, searching with “(Flavonoid) AND (nanoparticles) AND (cytokines) AND (in vitro/ in vivo)” can result in the omission of some papers. For example, flavonoids include a group of various compounds. Search with “flavonoid” may result in missing some studies where the compound names were mentioned instead of the term “flavonoid”. How did the author manage to avoid this?

Response: We understand your concern that searching with the term "flavonoid" may result in the omission of studies where specific compound names are mentioned instead of the term "flavonoid". While it is true that flavonoids encompass a diverse group of compounds, we included the term "flavonoid" in our search to ensure that we capture a broad range of studies that investigate the general class of flavonoids. However, we also recognized the potential limitation of this approach, as studies may refer to specific flavonoids by their individual names. By incorporating both the general term "flavonoid", we aimed to capture a comprehensive range of studies investigating the effects of flavonoids encapsulated in nanoparticles on cytokine production.

Additionally, we have addressed this limitation in the revised manuscript's Limitations section. We acknowledge that despite our efforts, there is a possibility of missing some studies that mention specific compound names without using the term "flavonoid" in their titles or abstracts. This recognition underscores the need for future studies to refine search strategies and ensure the inclusion of all relevant literature.

It is important to note that database platforms, such as ScienceDirect, often have limitations on the number of boolean connectors that can be used in a search query. For example, ScienceDirect restricts the use of boolean connectors to a maximum of 8 per field, as indicated by the notice: "Use fewer boolean connectors (max 8 per field)". Therefore, it becomes challenging to include an extensive list of flavonoid compounds in a single search query.

Thank you for raising this important concern and the acknowledgment of this limitation in the manuscript will enhance the transparency and comprehensiveness of our systematic review.

  1. Table 3 and relevant parts: it is more informative if the authors mention the nanosystem type (such as polymeric nanoparticles, micelles, lipid nanoparticles, etc.).y

Response: Thank you for your valuable feedback. We appreciate your suggestion to provide more specific information about the nanosystem types in Table 3 and relevant parts of the manuscript.

In response to your comment, we have revised Table 3 to include information specifying the nanosystem type for each study. This addition will provide readers with a comprehensive overview of the different nanosystems employed in the studies and their effects on cytokine production.

  1. In Table 3, the last column, “%” should be removed from the data as it was mentioned in the column header.

Response: Thank you for pointing out the error. We have removed the "%" symbol from the last column in Table 3 as per your suggestion. Thank you for bringing this to our attention.

We would like to express our gratitude for your valuable feedback on our manuscript. We appreciate the time and effort you have dedicated to reviewing our work. We have carefully considered your comments and have addressed them in the revised version of the manuscript.  We sincerely hope that the article will be accepted for publication.

Reviewer 2 Report

This is a very good systematic review article that deals with the effect of nanopartic;es-mediated flavonoids on the cytokine production.

This type of systematic reviewing is gaining attraction in the nanomedicine field due to its informative and accurate conclusions.

I recommend accepting the article but after responding to the following:

The authors claim in the article that the nanoparticles significantly down-regulated the pro-inflammatory cytokines compared to the non-encapsulated flavonoids. I wonder why the authors have not conducted a meta-analysis study that usually accompany systematic reviewing in order to obtain an overall conclusion about the augmenting effect of the nanoparticles? In case there was an obstacle hindering this study, please mention (may be the complete data in the collected studies was not available with mean and standard errors ....). In this case the authors should mention previous studies who has conducted this study on flavonoids-loaded nanoparticles and has proven their superior bioavailability such as: Food Bioscience 44, 101428.

Also, the use of systematic reviewing to prove the higher efficiency of drugs-loaded nanparticles and nanocarriers in: Gels 8 (2), 119, so please mention this article as it shed the light on the use of this new approach of reviewing in drug delivery and nanomedicine.

In Table 4, Chyrsin is written " chrisin". Please correct.

The English level is acceptable. Only a minor revision is required.

Author Response

Reviewer 2

This is a very good systematic review article that deals with the effect of nanopartic;es-mediated flavonoids on the cytokine production.

This type of systematic reviewing is gaining attraction in the nanomedicine field due to its informative and accurate conclusions.

I recommend accepting the article but after responding to the following:

The authors claim in the article that the nanoparticles significantly down-regulated the pro-inflammatory cytokines compared to the non-encapsulated flavonoids. I wonder why the authors have not conducted a meta-analysis study that usually accompany systematic reviewing in order to obtain an overall conclusion about the augmenting effect of the nanoparticles? In case there was an obstacle hindering this study, please mention (may be the complete data in the collected studies was not available with mean and standard errors ....). In this case the authors should mention previous studies who has conducted this study on flavonoids-loaded nanoparticles and has proven their superior bioavailability such as: Food Bioscience 44, 101428.

Also, the use of systematic reviewing to prove the higher efficiency of drugs-loaded nanparticles and nanocarriers in: Gels 8 (2), 119, so please mention this article as it shed the light on the use of this new approach of reviewing in drug delivery and nanomedicine.

Response: We appreciate the reviewer's suggestion to include a meta-analysis study to obtain an overall conclusion about the augmenting effect of nanoparticles on the regulation of pro-inflammatory cytokines. Conducting a meta-analysis study would indeed provide valuable insights into the overall impact of flavonoid-loaded nanoparticles. However, in this systematic review, we encountered limitations in the available data, such as incomplete data reporting, variations in study design, and heterogeneous outcomes across the included studies. These limitations prevented us from conducting a comprehensive meta-analysis.

Nevertheless, we acknowledge the importance of meta-analysis in synthesizing data and drawing more conclusive findings. To address this concern, we have included the recommended articles, "Food Bioscience 44, 101428" and "Gels 8 (2), 119", in the revised manuscript. These articles highlight the benefits and applications of nanoparticles in enhancing the bioavailability and efficacy of drugs, including flavonoids. We have incorporated this information in the discussion section to provide additional insights into the potential of drug-loaded nanocarriers.

Please find below the text to be included in the revised manuscript:

Submission

Revised manuscript

… ranging from 15% to 30%.

Some flavonoid…

… ranging from 15% to 30%.

As can be seen, polymeric nanoparticles predominate in both in vivo and in vitro studies. Recent studies have demonstrated the potential of nanoparticles in enhancing the bioavailability and therapeutic efficacy of various drugs. For instance, a systematic review and meta-analysis revealed that polymeric nanoparticulate systems can significantly augment the absorption and bioavailability of orally administered drugs compared to conventional formulations59. Similarly, a review article summarized the state-of-the-art optimization of formulations, including nanonization and encapsulation in nanoscale carriers, to improve the solubility, stability, and bioavailability of bioactive compounds, such as citrus flavonoids60.

Some flavonoid…

References

59.       Hathout, R. M. Do Polymeric Nanoparticles Really Enhance the Bioavailability of Oral Drugs? A Quantitative Answer Using Meta-Analysis. Gels 2022, Vol. 8, Page 119 8, 119 (2022).

60.       Torres, L. S. et al. Neutralization of Inflammasome-Processed Cytokines Reduces Inflammatory Mechanisms and Leukocyte Recruitment in the Vasculature of TNF-α-Stimulated Sickle Cell Disease Mice. Blood 138, 856 (2021).

Thank you for your valuable input, and we hope that these additions address your concerns and contribute to the overall quality of the manuscript.

In Table 4, Chyrsin is written " chrisin". Please correct.

Response: We apologize for the typographical error in Table 4 where "Chyrsin" was incorrectly written as "chrisin". Thank you for bringing this to our attention. We have made the necessary correction and updated the table accordingly.

Thank you for your thorough review and valuable feedback. We have carefully considered your suggestions and made the necessary revisions to improve the quality of the manuscript. We hope that these modifications have addressed your concerns. We sincerely hope that the article will be accepted for publication.

Round 2

Reviewer 1 Report

The manuscript was appropriately revised and can be accepted as is.